# Resting-State EEG Alterations of Practice-Related Spectral Activity and Connectivity Patterns in Depression

**DOI:** 10.3390/biomedicines12092054

**Published:** 2024-09-10

**Authors:** Elisa Tatti, Alessandra Cinti, Anna Serbina, Adalgisa Luciani, Giordano D’Urso, Alberto Cacciola, Angelo Quartarone, Maria Felice Ghilardi

**Affiliations:** 1Department of Molecular, Cellular & Biomedical Sciences, School of Medicine, City University of New York, New York, NY 10031, USA; alessandra.cinti@unifi.it (A.C.); aserbin000@citymail.cuny.edu (A.S.); adalgisaluciani@outlook.com (A.L.); 2Siena Brain Investigation & Neuromodulation Lab (Si-BIN Lab), Unit of Neurology & Clinical Neurophysiology, Department of Medicine, Surgery & Neuroscience, University of Siena, 53100 Siena, Italy; 3Department of Psychology, City College of New York, City University of New York, New York, NY 10031, USA; 4Department of Neurosciences, Reproductive and Odontostomatological Sciences, University of Naples “Federico II”, 80131 Naples, Italy; giordanodurso@gmail.com; 5Brain Mapping Lab, Department of Biomedical, Dental Sciences & Morphological and Functional Imaging, University of Messina, 98125 Messina, Italy; alberto.cacciola0@gmail.com; 6IRCCS Centro Neurolesi “Bonino Pulejo”, 98124 Messina, Italy

**Keywords:** depression, EEG oscillations, fractal dynamics, phase–amplitude coupling, resting-state EEG, beta frequency, gamma frequency, plasticity, energy

## Abstract

Background: Depression presents with altered energy regulation and neural plasticity. Previous electroencephalography (EEG) studies showed that practice in learning tasks increases power in beta range (13–30 Hz) in healthy subjects but not in those with impaired plasticity. Here, we ascertain whether depression presents with alterations of spectral activity and connectivity before and after a learning task. Methods: We used publicly available resting-state EEG recordings (64 electrodes) from 122 subjects. Based on Beck Depression Inventory (BDI) scores, they were assigned to either a high BDI (hBDI, BDI > 13, N = 46) or a control (CTL, BDI < 7, N = 75) group. We analyzed spectral activity, theta–beta, and theta–gamma phase–amplitude coupling (PAC) of EEG recorded at rest before and after a learning task. Results: At baseline, compared to CTL, hBDI exhibited greater power in beta over fronto-parietal regions and in gamma over the right parieto-occipital area. At post task, power increased in all frequency ranges only in CTL. Theta–beta and theta–gamma PAC were greater in hBDI at baseline but not after the task. Conclusions: The lack of substantial post-task growth of beta power in depressed subjects likely represents power saturation due to greater baseline values. We speculate that inhibitory/excitatory imbalance, altered plasticity mechanisms, and energy dysregulation present in depression may contribute to this phenomenon.

## 1. Introduction

Depression and anxiety represent major mental health concerns with their prevalence showing upward trends mostly in young subjects [1,2]. Although clinical diagnosis remains the purview of mental health professionals, several approaches and instruments have been used to identify biomarkers and predictors of depression as well as its response to therapies. Electroencephalography (EEG) recorded at rest is a particularly cost-effective and widely accessible method for functional brain imaging. Power analysis of EEG frequency bands in individuals with depression can yield insightful results by disclosing alterations that occur in specific bands and that may be associated with neural mechanisms underlying depressive symptoms. Notably, a debated EEG correlate of major depression is the frontal asymmetry in the alpha range, that is, the unequal activation of the two hemispheres [3] often associated with altered emotional processing and regulation. Moreover, different degrees and patterns of alterations in beta [4] and gamma [5] bands have been reported in depression.

A series of works now supports the hypothesis that depression can be described as a network disorder with particular emphasis on metabolic regulation, synaptic plasticity, and excitatory/inhibitory balance [6,7,8]. In this respect, there is accumulating evidence that depression is characterized by: (i) an imbalance of excitation and inhibition processes [9,10,11]; (ii) plasticity alterations [12,13,14]; and (iii) dysregulation of energy mechanisms [7], a dysfunction that has direct effects on both excitation/inhibition balance and plasticity. The impaired energy balance in depression has been substantiated by findings of brain lactate increases in these patients [15,16] and by the fact that lactate increases are coupled with mitochondrial dysfunction [17,18,19,20]. Direct evidence of elevated lactate levels in depressed patients has been reported in the ventricular CSF [21,22] and the anterior cingulate [23] using magnetic resonance spectroscopy (MRS). However, MRS’s limited availability and high costs preclude its use for extended studies and clinical application in depression. 

Recent evidence suggests that EEG beta frequency band may have a metabolic meaning, as its power magnitude directly correlates with high levels of cerebral lactate concentration [24,25] as well as with the energy mechanisms involved in long-term potentiation (LTP) processes. Indeed, we found in normal subjects that local levels of beta power during resting wake increase immediately after a visuo-motor learning task over the areas previously involved in the learning activity [26]. Importantly, such beta power increases vanish after either quiet wake or sleep [26,27], in agreement with the animal findings of lactate concentration decreases during quiet wake [24]. Further support for the association between energy dysregulation and beta power comes from studies in patients with Parkinson’s disease, showing that the levels of beta power are greater than in normal controls and do not increase with practice in a motor adaptation task [28]. Parkinson’s disease is a disorder that often presents with depressive symptomatology and is characterized by impaired LTP processes and energy dysregulation [29,30,31], similarly to depression. Therefore, we hypothesize that the EEG during resting states will show greater beta power in subjects with depression than in normal controls and display significant increases in beta power following practice of a task in controls but not in subjects with depression.

In this study, we took advantage of a publicly available dataset of EEG recorded during quiet wake in two groups of college students, one with normal Beck Depression Inventory (BDI) scores and the other with high BDI scores [32,33]. We analyzed the EEG recordings collected before and after performance in a probabilistic learning task.

## 2. Materials and Methods

### 2.1. Subjects, Task, and Experimental Design

The dataset used for this study was retrieved from the open-access data archive OpenNeuro (https://openneuro.org/datasets/ds003478/versions/1.1.0, accessed on 3 March 2023) [34] and included clinical data and raw resting EEG recordings from 122 college students [32,33]. Participants were recruited based on a mass survey with BDI [35] conducted at the University of Arizona. General criteria to enter the study were: (i) age between 18 and 25 years; (ii) no history of seizures or head trauma; and (iii) no current use of medications with psychoactive effects. Before the experimental procedure, subjects were clinically reassessed with the electronic mini-international neuropsychological interview and were classified as belonging to one of two groups. Participants without a history of major depressive disorders, BDI scores < 7 at both the initial and subsequent assessments, and without symptoms suggesting an Axis I disorder were included in the control group (CTL, N = 75, mean age: 19 ± 1.21; 40 women). Individuals with high BDI scores (>13) at the two testing points were included in the high BDI (hBDI) group (N = 46, mean age: 19 ± 1.14, 34 women). In addition to BDI, all participants completed the Spielberger Trait Anxiety Inventory (TAI) [36] and the Behavioral Inhibition Scale/Behavioral Activation Scale (BIS/BAS), a measure of reward and punishment reactivity [37].

EEG recordings were acquired at rest before and after a probabilistic learning task [32,33]. The probabilistic learning task is fully described by Cavanagh and colleagues [32,33]. Briefly, after the baseline EEG recording, participants completed two runs of a compulsory choice probabilistic learning task training and a testing phase [38]. During the training, three pairs of Japanese Hiragana characters were displayed; each pair of stimuli was linked to a probability of receiving a “Correct” or “Incorrect” feedback of 80/20%, 70/30%, or 60/40%. Each presentation lasted from 4300 to 4700 ms, and the stimuli disappeared upon the participant’s response. Feedback was provided 50 to 100 ms after the response during this training phase but not during the test. During the test phase, all possible combinations of stimuli pairs were presented eight times for a total of 120 trials.

As reported by Cavanagh and colleagues [32,33], the project and the consent forms were approved by the local IRB. Each participant provided written informed consent before entering the study.

### 2.2. EEG Data Acquisition and Preprocessing

Scalp EEG recordings were recorded for 6 min (3 min eyes open, EO, and 3 min eyes closed) using a Synamps2 system with 64 Ag/AgCl electrodes (band-pass filter: 0.5–100 Hz, sampling rate: 500 Hz, impedances < 10 kΩ). The online reference was a single electrode placed between Cz and CPz) [33]. EEG recordings were performed before and after the training and testing of probabilistic tasks. After downloading the publicly available raw data, we preprocessed the EEG signal only for the eyes open condition using the public MATLAB toolbox EEGLAB (version 2022) [39]. The continuous EEG signal was filtered using a finite impulse response (FIR) filter from 1 to 90 Hz and notch filtered from 56 to 64 Hz to remove power line noise (*pop_eegfiltnew*). The signal was then visually inspected to remove channels with poor signal quality. Independent component analysis (ICA) was run using the Infomax algorithm to identify eye blinks and horizontal eye movements, high-frequency muscular activity, heartbeat, and other periodic EEG artifacts (*pop_runica*). Component rejection was performed automatically using the EEGLAB toolbox IC Label, which automatically provides a proportion (in %) of independent components identified in EEG data that are classified into specific categories based on their characteristics [40]. Components labeled as Muscle (50–100%), Eye (80–100%), Heart (80–100%), and Line noise (80–100%) were rejected from the signal, resulting in an average of 36 ± 13.32 components kept per subject. Electrodes with bad signal quality were reconstructed using spherical spline interpolation (*pop_eeg_interp*), and the signal was re-referenced on the average channel activity (*pop_reref*). Subsequent analyses were carried out using custom data analysis scripts using the MATLAB-based Fieldtrip Toolbox (version 20220104) [41].

#### 2.2.1. Spectral Current Density Analyses

The preprocessed data was transformed by applying the Fieldtrip current source density (CSD) algorithm (*ft_scalpcurrentdensity*) to estimate the current flow underlying an EEG topography and reduce the impact of volume conduction on scalp-recorded data. After spatial filtering, the spectral profile for both the pre-task and post-task activity was estimated via fast-Fourier transform (multi-taper FFT with Hanning window) (from 1 to 90 Hz, 0.5 Hz bins, *ft_freqanalysis*). In addition to the “original” spectral power, we determined the distinct contributions of the oscillatory and fractal components. We thus decomposed the signal using irregular-resampled auto-spectral analysis (IRASA) [42]. This technique allows for the analysis of the “fractal background” (i.e., 1/f distribution of a time series) that reflects the background overall brain activity without specific rhythmic patterns. For the “original”, fractal, and oscillatory signals, statistical analyses were conducted on the theta (4–8 Hz), alpha (8.5–13 Hz), beta (13.5–25.5 Hz), gamma (30–90 Hz), low gamma (30–55 Hz), and high gamma (65–90 Hz) frequency ranges. We further parametrized the fractal, scale-free, and signal components by computing the power-law exponent (PLE). For each participant, the fractal component was thus transformed in log–log coordinates, and the distribution slope was computed for each channel using the MATLAB linear regression function *fitlm*, resulting in a single beta coefficient per channel.

#### 2.2.2. Phase–Amplitude Coupling (PAC)

In addition to the analysis of brain oscillatory activity and the distinct contributions of periodic and scale-free signals, we explored differences in functional connectivity to better understand how low-frequency and high-frequency oscillatory activity coordinate information processing in the brain. We first analyzed phase–amplitude coupling (PAC) between the phase of theta and the amplitudes of beta and gamma oscillations (*ft_crossfrequencyanalysis*) and computed the modulation index (MI) [43]. The selected bandwidth for both beta (11.5 Hz, ranging from 13.5 to 25 Hz) and gamma amplitude estimation (50 Hz, ranging from 30 to 80 Hz) exceeds the highest frequency within the theta phase band (4–8 Hz). This ensures a robust detection of PAC according to the criteria that the amplitude bandwidth should at least match the peak phase frequency. PAC estimation first involved filtering the signal using a two-way Butterworth filter and then its Hilbert transformation. As described by Tort and colleagues [43], we assessed PAC by dividing the phase (0–360°) into 18 segments and calculating the average amplitude for each segment, adjusted by the mean amplitude across all segments. As displayed in the formula below, the MI is derived by contrasting the observed amplitude-phase distribution (P) with a theoretical uniform PAC distribution (Q) using the Kullback–Leibler distance (D) grounded in the principles of Shannon’s entropy.
MI=D(P,Q)/log⁡(Nbins)

PAC values were computed individually for each subject, trial, and channel, and subsequently averaged to derive a single Modulation Index (MI) value per amplitude and phase for each channel. 

### 2.3. Statistical Analyses

#### 2.3.1. Clinical Measures

We used the following clinical outcome measures: BDI scores, including its anhedonia and melancholia subscales; STAI scores; BIS scores with the reward, fun-seeking, and drive subscales; and BAS scores. Differences between the hBDI and CTL groups were assessed with non-parametric independent samples using Mann–Whitney tests. Spearman correlation analyses were used to explore possible relationships between clinical scores, demographic variables, and EEG-derived measures. Finally, logistic regression analysis was performed to investigate whether BIS/BAS domains could serve as predictors of depressive symptomatology.

#### 2.3.2. Task Performance Accuracy

We obtained the task performance data from the first author of the original studies [32,33]. We averaged the mean accuracy for the two runs of the training and test phases to check for group differences with Mann–Whitney tests. Spearman correlation analyses were run for task accuracy and changes in resting-state EEG activity. 

#### 2.3.3. Spectral Current Density 

We first examined pre-task resting-state EEG activity to identify baseline differences between the two groups. Then, we conducted a similar comparison of post-task activity between the two groups. Finally, we investigated the impact of probabilistic learning practice on post-task activity in the two groups separately (post-task vs. pre-task). In all cases, we assessed whether significant differences were present from the original, periodic, and fractal components of the signal.

Statistical analyses on the original (or mixed) spectral power, the oscillatory, the scale-free components, and the extracted PLE values were computed using the non-parametric Monte Carlo-based permutation procedure as implemented in Fieldtrip [44]. At the first level of analysis, the null hypothesis that the two groups are equal (hBDI vs. CTL, independent *t*-test) was tested using a critical alpha of 0.05 and a minimum number of two neighboring electrodes to form a cluster. The second-level cluster-level statistic was computed using the sum of the t values within each cluster of electrodes. The maximum statistical value from the cluster-level analysis was finally compared using the Monte Carlo method with a distribution of maximum cluster values obtained after 10,000 permutations.

#### 2.3.4. PAC

Differences in theta–beta MI and theta–gamma MI strength between the hBDI and CTL groups, as well as changes with learning practice, were assessed using non-parametric cluster-based permutation testing (10,000 permutations, alpha = 0.05). As for the spectral analyses, initial comparisons were made between pre-task MI values across the two groups. Then, practice-related changes in functional connectivity (post vs. pre-task) within each group were evaluated, and subsequent comparisons examined post-task changes between the two groups.

## 3. Results

Age and test scores for the two groups are summarized in Table 1. As noted in the Methods, the hBDI group, which included a higher percentage of women (34 women out of 46 participants, 74%) than the CTL group (40 out of 75, 54%), had greater scores for BDI, TAI, and BIS. No significant group differences were observed for all BAS scores. We found no difference between men and women for all the clinical test scores.

BDI scores displayed a very strong positive correlation with TAI (ρ = 0.83, *p* < 0.001; confidence intervals (CI): 0.87, 0.75) and a moderate correlation with BIS scores (ρ = 0.49, *p* < 0.001; CI: 0.63, 0.32). TAI scores positively correlated with BIS scores (ρ = 0.61, *p* < 0.001; CI: 0.71, 0.48) and, to a lesser extent, negatively correlated with BAS scores (ρ = −0.18, *p* = 0.047; CI: −0.00083, −0.35).

Logistic regression analysis showed that the BIS score was a good predictor of acute depressive symptomatology (*X*^2^ = 27.51, *p* < 0.001; Wald stat = 20.24, *p* < 0.001; AUC = 0.77; Accuracy = 0.75; Sensitivity = 0.56; Specificity = 0.87), with greater sensitivity when the analysis was restricted to women (CTL: N = 39; hBDI: N = 33; *X*^2^ = 18.65, *p* < 0.001; *W* = 13.49, *p* = 0.0002; AUC = 0.77; Accuracy = 0.74; Sensitivity = 0.70; Specificity = 0.77).

### 3.1. At Baseline, hBDI Displays Greater Beta and Gamma Oscillatory Activity than CTL

We first analyzed the EEG collected at baseline before the probabilistic learning task. Cluster-based permutation statistics on the “original” spectral activity highlighted significant differences between the two groups in both beta and gamma frequency ranges (Figure 1, first column, see also Appendix A). Specifically, compared to the CTL, the hBDI group displayed greater beta power in a cluster of electrodes located over the frontal region (Cluster *t* = 25.81, *p* = 0.028) and greater gamma power in two clusters, one over the central midline (Cluster *t* = 23.70, *p* = 0.021) and the other one over the occipital region (Cluster *t* = 21.14, *p* = 0.023). No significant group differences were observed for the other frequency ranges. Analyses of the individual beta and gamma frequency peaks in the significant clusters were similar in the two groups (beta, hBDI: 16.53 ± 3.51 Hz, CTL: 16.84 ± 3.30 Hz; *W* = 1753, *p* = 0.45; gamma, hBDI: 31.98 ± 5.53 Hz, CTL: 32.83 ± 7.89 Hz, *W* = 1823, *p* = 0.23), suggesting that the greater beta and gamma activity we found in hBDI is not due to a shift in frequency peaks.

We then focused on the two EEG components obtained with IRASA. The analysis of the pure oscillatory component isolated from the fractal (Figure 1, second column, see also Appendix A) confirmed greater beta oscillatory activity over fronto-parietal areas in the hBDI group (Cluster *t* = 125.41, *p* = 0.004), as well as greater gamma activity over the right parieto-occipital region (Cluster *t* = 19.98, *p* = 0.023). Furthermore, the hBDI group displayed lower alpha activity in a cluster of electrodes over the right frontal (Cluster *t*= −27.76, *p* = 0.016) and left temporal (Cluster 2: *t* = −6.31, *p* = 0.040) scalp regions.

Regarding the fractal component, group differences were found only in the gamma range (Cluster *t* = 74.22, *p* = 0.003), with greater values for the hBDI group in midline and parieto-occipital electrodes (Figure 1, third column, see also Appendix A). Cluster-based analysis of the PLE index highlighted a significant group difference bilaterally over the parieto-occipital area (Cluster *t* = 24.18, *p* = 0.03), with hBDI having a reduced slope compared to the CTL (Figure 2A).

In summary, we found that the baseline EEG of the hBDI group was characterized by greater activity in the high frequency ranges (beta and gamma) than the CTL’s, mostly over fronto-parietal regions, with minor discrepancies between the original oscillatory, pure oscillatory, and fractal components. We did not find any significant correlation between original, oscillatory, and fractal components and the clinical measures for each of the two groups.

### 3.2. The hBDI Group Displayed Greater Theta-Beta and Theta-Gamma PAC 

We then analyzed PAC for theta–beta and theta–gamma by extracting the corresponding MI in the baseline EEG recordings. Non-parametric permutation analyses demonstrated increased theta–beta coupling mostly over a centro-parietal cluster (*t* = 23.16; *p* = 0.0175) (Figure 3A, first line) and greater theta–gamma coupling over the right occipital areas (*t* = 10.12; *p* = 0.048) in the hBDI group compared to the CTL group (Figure 3B, first line).

### 3.3. EEG Activity at Rest Changes after One-Hour Task Practice

After the baseline EEG, all subjects underwent two blocks of training and testing in a probabilistic learning task. The performance of the two groups was similar in both the training (CTL: mean ± SD: 68 + 9%; hBDI: 68 ± 10%; Mann-Whitney *U* = 1755, *p* = 0.87) and the test (CTL: 65 ± 9%; hBDI: 67 ± 10%; *U* = 1776, *p* = 0.79).

Cluster-based analysis of the “original” spectral activity following the task revealed lower power values in the low-frequency ranges, i.e., theta and alpha in hBDI compared to CTL (Figure 4, first column, see also Appendix A). The post-task result differs from the pre-task finding of local increases in the hBDI group for the higher frequency ranges, i.e., beta and gamma (see Figure 1, first column). Specifically, in the post-task recordings, compared to CTL, the hBDI group showed less theta range activity over a centro-parietal region (Cluster *t* = −12.87, *p* = 0.028) and in alpha range in a cluster of electrodes over the frontal region (Cluster *t* = −16.72, *p* = 0.025) (Figure 4, first column). There was no group difference for the peak frequency of theta (CTL: 5.82 ± 1.48 Hz; hBDI: 5.83 ± 1.44 Hz; *W* = 1568, *p* = 0.863) and alpha (CTL: 9.98 ± 1.01 Hz; hBDI: 10.33 ± 1.18Hz; *W* = 1288, *p* = 0.077) for the electrodes included in the clusters. 

We then ascertained whether this change of pattern was present for the oscillatory and fractal components in the post-task recordings. While we found no significant group differences for the fractal component (Figure 4, third column, see also Appendix A) or for the PLE (Figure 2B), the analysis of the oscillatory component confirmed that the hBDI group showed lower alpha power in a cluster of electrodes over the frontal region extending to the left parietal area (Cluster *t* = −46.49, *p* = 0.014) (Figure 4, second column, see also Appendix A). Nevertheless, the hBDI group showed greater beta power in electrodes over the frontal region (Cluster *t* = 40.28, *p* = 0.017) and, to a smaller extent, over the left parieto-occipital area (Cluster *t* = 12.53, *p* = 0.0497) (Figure 4, second column).

Altogether, the analyses of the post-task original and the pure oscillatory components produced results diverging from the pre-task findings in terms of group differences. Specifically, for the original oscillatory component, visual inspection of the pre- and post-task group differences (Figure 1 and Figure 4, see Table 2) suggests that task practice generated a more substantial power increase in the CTL than the hBDI group for all frequencies. For the pure oscillatory component, the group difference for the beta range fell in the post-task EEG in terms of both power amplitude and cluster magnitude (first columns of both Figure 1 and Figure 4) with the opposite effect for alpha frequency. Together with the absence of group differences in the gamma range in the post-task recordings, these findings support the hypothesis that, despite similar performance accuracy, the task activity induced different patterns of oscillatory changes in the two groups. To verify this hypothesis, we thus searched for post-pre task EEG changes in each of the two groups separately.

Cluster-based analyses of the original spectral activity in the CTL group confirmed that significant post-pre task increases occurred in all frequency ranges (Figure 5A, first column). Specifically, we found increased activity in (i) theta band: in two clusters of electrodes over frontal (Cluster *t* = 59.375, *p* = 0.003) and occipital areas (Cluster *t* = 50.32, *p* = 0.003); (ii) alpha band: in two clusters over the frontal (Cluster *t* = 64.04, *p* = 0.003) and right parietal areas (Cluster *t* = 19.44, *p* = 0.017); (iii) beta band: in a cluster of electrodes extending from the prefrontal to the occipital areas (Cluster *t* = 95.83, *p* = 0.0005); and (iv) gamma band: in two clusters over frontal (Cluster *t* = 45.63, *p* = 0.0006) and occipital areas (Cluster *t* = −65.73, *p* = 0.0004). 

The same comparison in the hBDI group displayed a different pattern with increases that affected only the theta and gamma ranges (Figure 5B, first column). Indeed, following the task, increased oscillatory was present for theta in a cluster of electrodes over the frontal region (Cluster *t* = 15.30, *p* = 0.019) and a smaller one over the occipital region (Cluster *t* = 7.70, *p* = 0.050) and for gamma in two clusters over the centro-parietal areas (right: Cluster *t* = 11.22, *p* = 0.023; left: Cluster *t* = 18.076, *p* = 0.010).

Post-pre task comparison of the pure oscillatory component (Figure 5A, second column) showed that, following task practice, the CTL group exhibited increased alpha activity in two clusters of electrodes, one over the frontal area (Cluster *t* = 28.38, *p* = 0.015) and the other over the right parieto-occipital region (Cluster *t* = 13.50, *p* = 0.034). Increased power was also found in the beta range over the central region (Cluster *t* = 13.20, *p* = 0.026) and in the gamma range in two clusters of electrodes, one over the frontal area (Cluster *t* = 39.73, *p* = 0.0006) and the other over the occipital region (Cluster *t* = 49.28, *p* = 0.0006). The same analyses performed in the hBDI group (Figure 5B, second column) revealed a different pattern of changes. Following the task, beta power decreased in a cluster of electrodes over the right parieto-occipital region (Cluster *t* = −20.87, *p* = 0.003) and gamma power increased in two clusters of electrodes over the fronto-centro-temporal areas (left: Cluster *t* = 15.68, *p* = 0.014; right: Cluster *t* = 12.92, *p* = 0.018).

For the fractal component, compared to pre-task EEG, CTL post-task recordings demonstrated a general power increase for all frequency ranges in almost all electrodes (Figure 5A, third column). Namely, increases were found in a big cluster involving most of the electrodes for theta (Cluster *t* = 130.57, *p* = 0.00001), alpha (Cluster *t* = 148.82, *p* = 0.00001), and gamma (Cluster *t* = 142.28, *p* = 0.00001), with beta power increasing in both a frontal (Cluster *t* = 72.91, *p* = 0.0024) and an occipital (Cluster *t* = 66.98, *p* = 0.0026) cluster. A significant post-pre difference in the PLE was found for all electrodes (Cluster *t* = 267.79, *p* < 0.0001). Post-task increases of the fractal component were less evident in the hBDI group (Figure 5B, third column) and did not involve the beta range. Theta increases were present in two small clusters of electrodes, one over the frontal area (Cluster t= 24.88, *p* = 0.00001) and another over the left occipital area (Cluster *t* = 8.32; *p* = 0.042); Alpha power increased in a small right frontal cluster (Cluster *t* = 14.56, *p* = 0.026) and gamma over the left centro-parietal area (Cluster *t* = 14.74, *p* = 0.014) and the right fronto-central (Cluster *t* = 11.40, *p* = 0.024) region. As for the CTL group, PLE increased significantly for all electrodes (Cluster *t* = 218.57, *p* < 0.0001).

In summary, the separate comparisons of the post-task and pre-task recordings in each group (Table 3) confirmed that practice in a probabilistic learning task induced a different pattern of EEG changes in both the CTL and hBDI, even though the two groups performed the task with similar accuracy. These last analyses highlighted that post-task changes involved increases in all frequency ranges for the CTL group but not for the hBDI group. Indeed, beta power increase was found only in CTL, while the hBDI displayed either no changes or even some power decrements of the pure oscillatory component.

### 3.4. Phase–Amplitude Coupling Is Greater in hBDI Only at Baseline 

Lastly, we analyzed group differences for theta–beta and theta–gamma PAC in the post-task recordings. Non-parametric permutation analyses demonstrated that, differently from the pre-task recordings, there were no group differences for theta–beta and theta–gamma MI in the post-task recordings (Figure 3B). Thus, in line with the results of post-task original and oscillatory components, these results suggest that task activity had differential effects on the MI of the two groups, despite similar performance rates. Indeed, separate group analyses (Figure 6) revealed that, in the post-task compared to the pre-task recordings, theta–beta MI of the CTL group increased in electrodes over a right fronto-central region (Cluster *t* = 22.63, *p* = 0.0018, SD: 0.0004, CI: 0.0008) and the left central area (Cluster *t* = 11.63, *p* = 0.014, SD: 0.0012, CI: 0.0023) (Figure 6A, first line). Theta–gamma PAC increased in most electrodes (Cluster *t* = 114.74, *p* = 0.00001, SD: 0.0012, CI: 0.0023) (Figure 6B, first line). The same comparisons in the hBDI group did not show any significant change for theta–beta PAC (Figure 6A, bottom line), while theta–gamma MI increased over an extended area on both hemispheres without involvement of the mid-frontal electrodes (Cluster *t* = 80.56, *p* = 0.0005, SD: 0.0002, CI: 0.0004) (Figure 6B, bottom line). These findings suggest that substantial and extensive increases of both theta–beta and theta–gamma PAC occurred post-task in CTL, while no changes were noted in hBDI for theta–beta PAC.

## 4. Discussion

In this study, we analyzed EEG results recorded during resting states before and after the performance of a learning task in a large cohort of young college students who were tested for depression and anxiety. As summarized in Table 2, we found that, compared to CTL, subjects with high BDI scores showed greater power in the high-frequency range, beta and gamma, mostly over fronto-parietal regions. However, after performing a learning task with similar accuracy, such differences either disappeared or diminished (see Table 2 and Table 3). Specifically, post-task EEG changes showed substantial increases in all frequency ranges (with a notable increase in beta) for the CTL group, but not for the hBDI group. Further analyses focused on the pure oscillatory and fractal components as well as on theta–beta and theta–gamma PAC revealed additional differences between the two groups, both before and after the learning task. In the following paragraphs, we will first discuss the characteristics of our study population and then explore the potential implications and meanings of the EEG differences (in terms of oscillatory and fractal components) observed between the CTL and hBDI groups at baseline and their changes following the learning task. We will then comment on our findings about theta–beta and theta–gamma PAC at baseline and following the task.

### 4.1. Over-Sensitive Behavioral Inhibition System Is Linked to Depression and Anxiety

The demographic characteristics of the hBDI group were similar to those reported in the 2021 NSDUH Annual National Report at https://www.samhsa.gov/data/report/2021-nsduh-annual-national-report (accessed on 12 April 2024). In fact, compared to CTL, our depressed cohort included more women than men and had higher levels of trait anxiety, sensitivity to punishment, and withdrawal from aversive or unfamiliar stimuli. BAS displayed a weak negative correlation to trait anxiety scores but not to the BDI scores. The association of anxiety with depression and the correlations between clinical tests suggest that all these psychological constructs are linked [45,46,47,48]. These findings are in line with Gray’s reinforcement sensitivity theory about the role of personality in the genesis of mood and anxiety disorders [49,50,51]. Indeed, increased behavioral inhibition sensitivity may be a risk factor for affective disorders, with higher BIS scores linked to elevated levels of trait anxiety and depression [52]. Despite the small sample, our results are in agreement with previous accounts [53,54] that depression may be associated with an overactive inhibitory system and a hypoactive activation system, while anxiety may be more related to heightened inhibitory sensitivity and avoidance behavior [49]. The results of our logistic regression model showing that BIS accurately identified 70% of individuals with high BDI scores further strengthens the value of BIS as a good predictor of an ongoing depressive status.

### 4.2. Depression Is Characterized by Greater Beta Power during Resting-State EEG before the Task

In line with previous reports [4], the group with depressive symptoms displayed greater beta oscillatory activity compared to CTL over the fronto-parietal regions in the resting-state EEG results recorded before the task. Frontal beta activity has been associated with top-down control of attention-related mechanisms and emotional processing [55,56]. Studies in normal and depressed subjects have reported that beta power over frontal areas increases with attention and working memory [57,58], stress [57,59], and anxiety [60], and decreases with inattention [61]. Executive dysfunction and attentional deficits, such as impaired concentration and decision-making abilities, are indeed common in depression [62,63,64,65]. The role of beta in executive functioning is also supported by analyses of phase synchronization [55,66] and theta–beta ratio (an index reflecting attentional control, top-down inhibition, and cognitive processing [66,67]).

What is the meaning of greater frontal beta power in depression? As shown by MRI studies [68,69], the frontal areas may be particularly active in depressed subjects to maintain a normal level of performance during working memory tasks. Accordingly, the frontal enhancement of beta oscillations in the hBDI group could be interpreted in the frame of compensatory mechanisms to increase the efficiency of attentional control and maintain cognitive and emotional functioning. On the other hand, it may reflect hyperarousal linked to anxiety and stress [70]. While plausible, these conclusions remain, at least for the moment, hypotheses to be tested in ad-hoc studies, as the database we used did not include cognitive functioning assessment.

The observed beta power increase could also be linked to power decrements in the lower frequency ranges, such as the lower alpha oscillatory activity we found over the right frontal and left temporal areas. Decreased frontal alpha activity is a well-documented finding in depression [3,71]. Frontal alpha asymmetry in depressive disorders was first described in 1983 [72], with the left frontal area displaying higher alpha band power than the right one [71,72,73,74,75]. Although some publications have now questioned it, frontal alpha asymmetry has been mainly linked to emotional dysregulation and other factors such as motivation and positive affect (for a review, see: [3,76]), with reduced right frontal alpha reflecting negative emotional states and withdrawal [3,71,77].

Another interesting finding is that increased beta power over the frontal area and the reduced alpha power in hBDI compared to controls were present only for the mixed and pure periodic component of the signal but not for the fractal component. Previously discarded as “1/f noise”, the scale-free dynamics are thought to be generated by mechanisms that differ from those ruling periodic oscillatory activity and have been hypothesized to reflect neural complexity [42,78,79]. Thus, the sole increase in the oscillatory component we observed in the hBDI group indicates that such changes likely reflect localized brain functions, such as heightened cognitive or emotional processing, that rely on a specific, state-dependent, rhythmic activity.

### 4.3. Gamma Power at Rest Is Greater in Depression 

Subjects in the hBDI group also displayed greater parieto-occipital gamma power, a finding that could further support the association between depression and altered attentional processes. Lately proposed as a putative biomarker of depression, studies on the role of gamma oscillatory activity in depression have yielded mixed results, with some of them reporting reductions and others reporting increases in gamma activity [5]. These discrepant results may stem from differences in pharmacological treatments: serotonergic drugs (e.g., fluoxetine) lower gamma activity, while noradrenergic drugs (e.g., reboxetine) increase it [5,80,81]. In this line, other investigations have suggested that the combination of increased cortical levels of glutamate and reduced GABAergic activity is a biomarker of trait vulnerability to depression rather than a correlate of mood alterations [82,83], although such excitatory and inhibitory imbalances have been often associated with depression [9,82,83,84,85]. The link between increased glutamatergic activity and high-frequency oscillatory activity has been further confirmed by a model showing that slowing down glutamate decay increases both beta and gamma oscillatory activity [8]. Therefore, our results of increased high-frequency oscillatory activity could reflect altered glutamatergic signaling [86].

It should be noted that the power increase in the gamma range in the hBDI group also involved the fractal component, suggesting a different or disrupted dynamic complexity in the hBDI group. Moreover, our finding of a flatter broadband power-law decay (PLE) over the parieto-occipital areas in hBDI parallels previous findings of reduced slope in individuals with trait anxiety [87]. Similar results have also been reported during cognitive performance [88,89] as well as in aging and dementia [90], further supporting associations between depression, altered cognitive processes, and abnormal power-law scaling. As a steeper broadband spectral slope suggests more efficient information processing [91], the flatter slope observed in our hBDI sample may be interpreted as a biomarker of increased neural noise and dysregulated brain function. While additional studies are needed to confirm our interpretations, these findings underscore the importance of examining both periodic and aperiodic components in brain activity to get a comprehensive picture of the neural alterations that occur in and can predict depression.

### 4.4. Phase–Amplitude Coupling Is Greater in hBDI Only before the Task

We found increased theta–beta PAC over the parietal region and theta–gamma coupling over the right occipital area in our hBDI subjects. From a behavioral point of view, increased PAC, particularly between theta and gamma oscillations, has been linked to various cognitive processes, such as working memory, attention, and cognitive flexibility, abnormalities often found in depression. Previous studies on depression have reported conflicting results about PAC, with some showing increased coupling [92,93], as in our case, and others showing decreased coupling in different frequency ranges [94,95]. Such contrasting results might be due to differences in study designs and conditions tested, patient sample characteristics, specific brain regions examined, and other factors including time of testing. For example, intracranial recordings in epileptic patients showed that low/middle to high-frequency PAC was increased in clinically depressed subjects with epilepsy compared to subjects with epilepsy only, with greater PAC values found in the evening/night than during the daytime [92]. Also, a study of untreated patients at their first depressive episode found greater resting theta–gamma PAC in depressed patients compared to controls [93]. In that case, recordings were performed at rest and during an auditory test with alternating baseline and stimulation periods.

An important finding of the present work was that, after the task, the hBDI group showed some increase only in the theta–gamma PAC while the control group displayed a substantial increase in both theta–beta and theta–gamma PAC. As a result, the group difference found at baseline for both theta–beta and theta–gamma PAC basically vanished following the learning task. While from a behavioral point of view it may underlie cognitive aspects, low-high frequency PAC may be, at the system level, a measure of information processing and facilitation within and between cortical networks [96] that depends on a proper balance of excitatory and inhibition mechanisms at the cellular level [97,98,99,100,101,102]. Consequently, PAC plays an important role in long- and short-term potentiation processes, as shown also by studies in animal models of depression [97]. Thus, the lack of substantial PAC increase in the hBDI group after the task may be an expression of decreased plasticity, a finding previously described in depression [13]. In turn, altered plasticity may be linked to brain energy dysregulation that has been reported in this disease [7,103,104,105] as discussed at the end of next section. 

### 4.5. Oscillatory Activity Increases after Task Practice in CTL but Not in hBDI 

The analyses of the EEG recordings after the task revealed a different pattern of changes in the two groups, despite almost identical performance scores in the task. In general, the post-task resting-state EEG of the CTL group was characterized by broadband power increases in alpha, beta, and gamma ranges of both the pure oscillatory and fractal components and in theta range for the fractal component only. These changes of the oscillatory component are in line with our previous findings that after practicing learning tasks, the resting-state EEG of healthy subjects displays local increases of power mostly in beta and theta ranges [26] and also in gamma [106], that occur mostly over the areas previously engaged in the task. Conversely, in the present hBDI group, we found a decrement of beta power in a few electrodes over the parieto-occipital area and a modest increase in gamma power involving fewer electrodes than the CTL group. The greater baseline oscillatory activity of the hBDI group may impede further growth because the power may have reached saturation levels. In previous work, we suggested that power increases found after task performance may represent the electrical signature of metabolic processes related to short and long-term potentiation induced by learning and practice [25,26,106]. We thus speculate that the lack of substantial EEG changes following task practice in the hBDI group together with our findings on theta–beta PAC (see previous section) may reflect a defective engagement of plasticity-related processes that include excitatory/inhibition balance and energy regulation. This conclusion is supported by three sets of considerations: 1. plasticity-related processes, excitatory/inhibition balance, and energy regulation are impaired in depression [103,107]; 2. the changes in beta may signal lactate availability needed to produce fast energy resources for brain activity [24,25], and the changes in gamma may be related to mitochondrial activity [108]; 3. there is direct evidence of mitochondrial dysfunction and increased lactate levels in the brains of depressed patients [21,22,23] and that lactate increases are coupled with mitochondrial dysfunction [17,18,19,20]. Indeed, we acknowledge the need for targeted studies to prove our hypothesis that practice-related beta and gamma power changes can be biomarkers of energy dysregulation in depression. Such investigations would be rather important, especially in the light of recent studies proposing to use lactate as an antidepressant drug [107].

### 4.6. Limitations

The present study has several limitations. A good number of them are related to the dataset we chose because of its great advantage of EEG recordings before and after a task in subjects with high and low BDI scores. Specifically, this dataset did not provide information about the socioeconomic status of the participants, did not include participants with a BDI score between 7 and 16 (thus precluding analysis with continuous BDI scores), and only included clinical scores of BDI, STAI, BIS, and BAS. Such scores were largely intercorrelated; individuals of the hBDI groups also displayed greater trait anxiety compared to CTL, making it very difficult to disentangle the contribution of anxiety to EEG abnormalities without additional clinical measures. Most importantly, the dataset did not offer a comprehensive assessment and testing of aspects related to cognition and plasticity mechanisms that may be part of depression.

Despite the clear group differences in EEG-derived measures, we did not find significant correlations between them and the clinical scores, suggesting that the observed spectral and connectivity differences might be explained by other psychological and cognitive factors. Among them, attentional and working memory abnormalities, stress, and sleep quality are important factors for plasticity and often accompany depression. The lack of this type of information severely curtails the interpretation of our EEG findings, especially as biomarkers of specific factors associated with depression.

Another limitation of this investigation is that subjects were all college students within a narrow age range. This aspect likely decreased variability but makes it critical to validate the present results for other age ranges and populations with stressors other than those related to specific life experiences. Therefore, caution should be exercised in extending these results to older people or to populations from different socioeconomic and cultural backgrounds, and future investigations are needed to address these issues. Moreover, it would be important to replicate the present findings using other types of tasks, which may induce different topographical patterns, to assess the generalizability of our conclusions in terms of frequency changes. Finally, among other limitations, there is the lack of information about previous antidepressant treatments and major depression episodes. As discussed in the previous paragraphs, these factors may influence brain inhibition/excitatory balance and thus EEG results.

## 5. Conclusions

This is the first study demonstrating that in depressed subjects, a period of practice in a learning task does not induce the same increases observed in healthy individuals of higher frequency oscillatory activity recorded during resting-state EEG. This deficit likely reflects alterations of plasticity-related mechanisms and brain energetic metabolism. However, only future EEG studies coupled with techniques specifically exploring brain metabolism, such as MRS, will provide a comprehensive picture of the dysfunction underlying depression and its symptoms. These studies, in turn, will help foster the development of novel and targeted therapies based on pharmacological and non-pharmacological approaches.

## Figures and Tables

**Figure 1 biomedicines-12-02054-f001:**
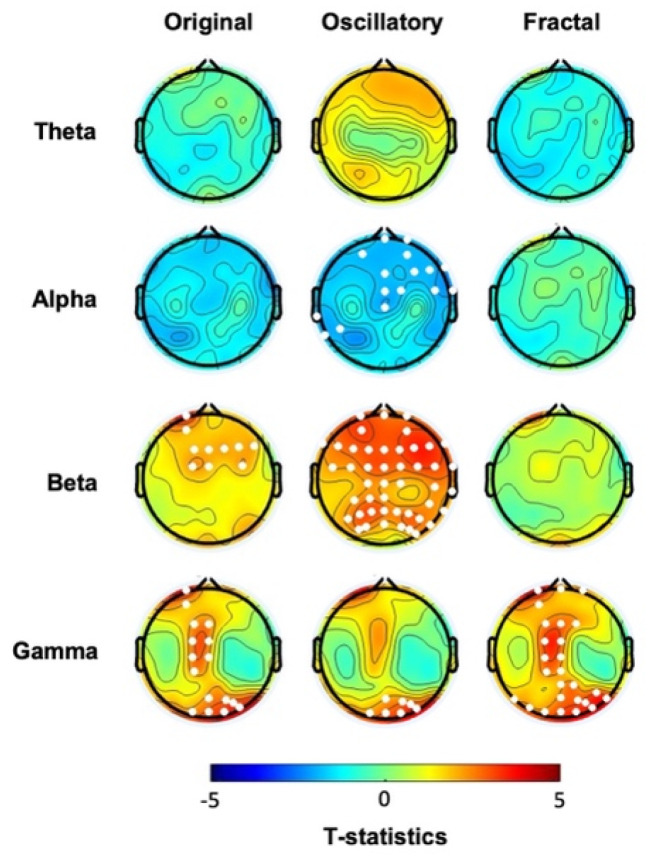
Scalp topographies displaying the *t*-values for group comparisons (hBDI vs. CLT) before the task, at baseline for the original, oscillatory, and fractal components for each frequency band. White dots indicate electrodes with significant group differences after cluster correction for multiple comparisons.

**Figure 2 biomedicines-12-02054-f002:**
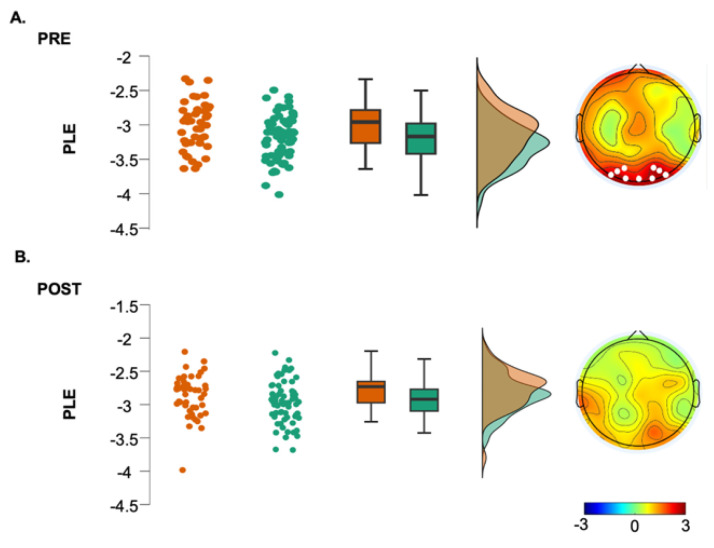
EEG Power-Law Exponent (PLE) for the hBDI group (orange) and CTL group (green) before (**A**) and after the task (**B**), with individual data points, box plots, and density plots to visualize data distribution. Topographic plots depict the *t*-values of cluster-based permutation statistics (right). Electrodes within significant clusters are represented as white dots.

**Figure 3 biomedicines-12-02054-f003:**
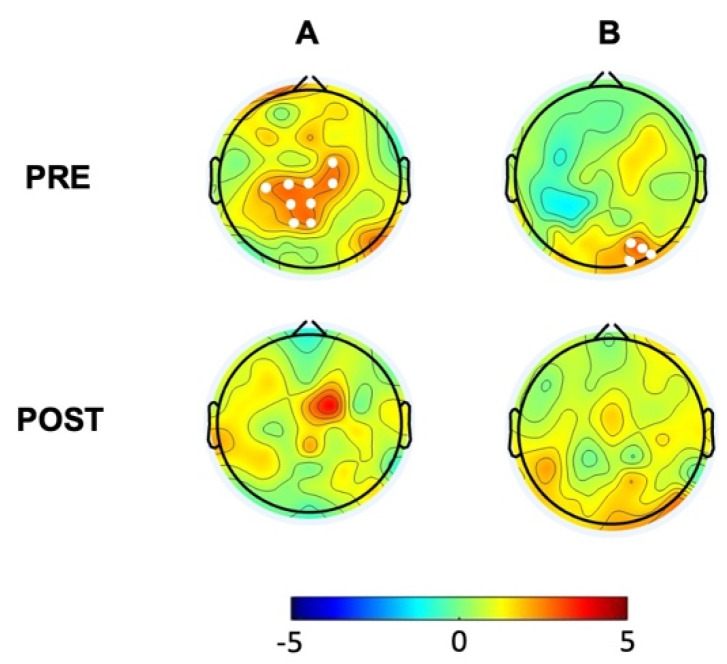
Group differences (hBDI vs. CTL) for theta–beta (**A**) and theta–gamma (**B**) PAC. Topographic plots depict the results of cluster-based permutation *t*-statistics. PRE, in the top line, represents the group comparison at baseline, before task performance. POST (bottom line) refers to group comparison after the task execution. White dots represent electrodes within significant clusters.

**Figure 4 biomedicines-12-02054-f004:**
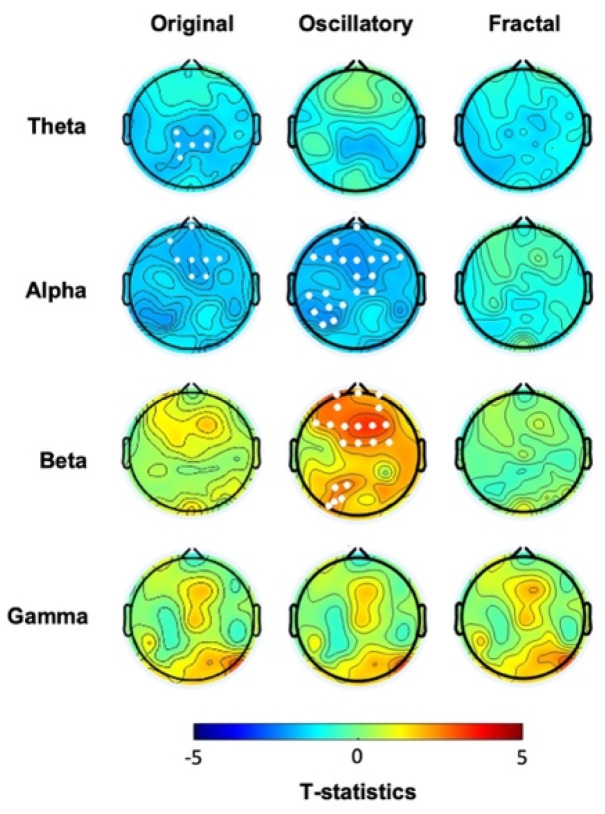
Scalp topographical *t*-maps of group comparisons (hBDI vs. CLT) in the post-task resting-state EEG for the original, oscillatory, and fractal components. White dots indicate electrodes with significant group differences after cluster correction for multiple comparisons.

**Figure 5 biomedicines-12-02054-f005:**
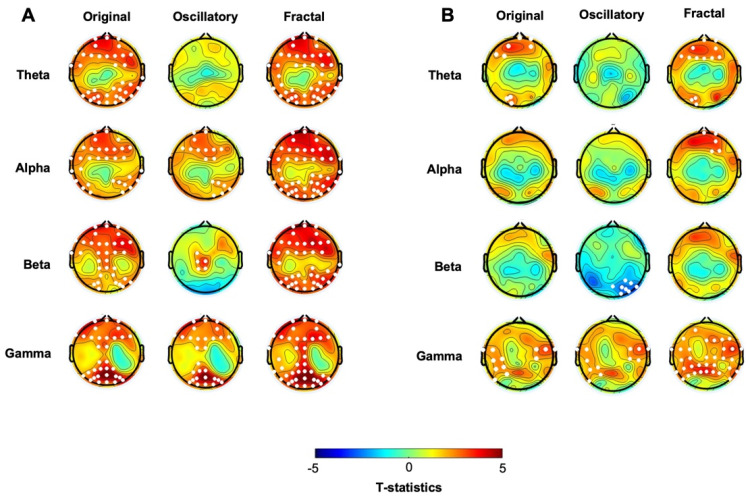
Scalp topographical *t*-values maps for post-pre task comparisons (post-task vs. pre-task EEG) in the CTL (**A**) and hBDI (**B**) groups. Results for the original, oscillatory, and fractal components are presented for each frequency band. White dots indicate electrodes with significant group differences after cluster correction for multiple comparisons.

**Figure 6 biomedicines-12-02054-f006:**
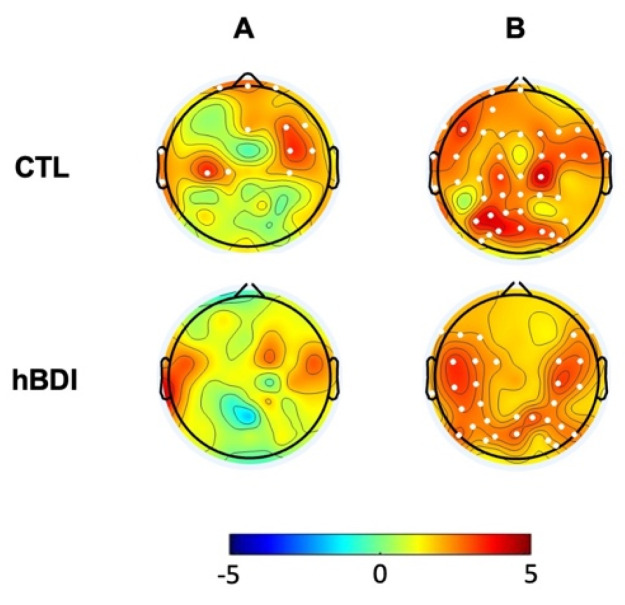
Post-task/pre-task differences for theta–beta (**A**) and theta–gamma (**B**) PAC in the CTL (first line) and hBDI (bottom line) groups. Topographic plots depict the results of cluster-based permutation *t*-statistics. White dots represent electrodes within significant clusters.

**Table 1 biomedicines-12-02054-t001:** Mean of age and scores of neuropsychological tests of the control (CTL) and depressed (hBDI) groups. In parentheses we report the standard deviation (SD). The values of *U* Mann-Whitney tests comparing the two groups are reported with *p*-values. Significant group differences are reported in bold. (BDI: Beck Depression Inventory; TAI: Trait Anxiety Inventory; BIS: Behavioral Inhibition System; BAS: Behavioral Activation System; BAStot: BAS total score; BASrew: BAS reward score; BASfun: BAS fun-seeking score; BASdrv: BAS drive score).

	CTL Mean (SD)	hBDI Mean (SD)	*U*	*p*-Value
Age (yrs)	18.97 (1.22)	18.74 (1.14)	1.30	0.25
BDI	**1.73 (1.65)**	**22.22 (4.9)**	**85.90**	**<0.001**
TAI	**31.05 (5.49)**	**55.76 (7.08)**	**83.49**	**<0.001**
BIS	**19.44 (3.05)**	**22.72 (3.07)**	**26.38**	**<0.001**
BAStot	40.45 (4.94)	39.17 (5.68)	1.07	0.30
BASrew	17.67 (1.71)	17.13 (2.15)	1.52	0.22
BASfun	11.88 (2.17)	11.52 (2.93)	0.07	0.79
BASdrv	10.89 (2.15)	10.52 (2.79)	0.03	0.86

**Table 2 biomedicines-12-02054-t002:** Summary of group differences (hBDI > CLT) of the original (ORIG), pure oscillatory (OSCIL), and fractal (FRACT) activity for theta (4–8 Hz), alpha (8.5–13 Hz), beta (13.5–25.5 Hz), and gamma (30–90 Hz) before (Pre-task) and after (Post-task) the learning task. In red, we report results where the hBDI group exhibited significantly greater activity than CTL. In blue, we report results where CTL displayed significant activity. “None” refers to no significant group differences. R = right; L = left.

	Pre-Task: hBDI > CTL	Post-Task: hBDI > CTL
Theta	ORIG: *none* OSCIL: *none* FRACT: *none*	**ORIG: ***centro-parietal* OSCIL: *none* FRACT: *none*
Alpha	ORIG: *none* **OSCIL: ***R frontal & L temporal* FRACT: *none*	**ORIG: ***frontal* **OSCIL: ***from frontal to L parietal* FRACT: *none*
Beta	**ORIG: ***frontal* **OSCIL: ***fronto-parietal* FRACT: *none*	**ORIG: ***frontal* **OSCIL: ***fronto-parietal* FRACT: *none*
Gamma	**ORIG**: *central midline & occipital* **OSCIL**: *R parieto-occipital* **FRACT**: *midline & parieto-occipital*	**ORIG**: *central midline & occipital* **OSCIL**: *R parieto-occipital* FRACT: *none*

**Table 3 biomedicines-12-02054-t003:** Summary of the differences between post-task and pre-task for the original (ORIG), pure oscillatory (OSCIL), fractal (FRACT) EEG activity in the CTL and the hBDI groups for theta (4–8 Hz), alpha (8.5–13 Hz), beta (13.5–25.5 Hz), and gamma (30–90 Hz). In red, we report results where post-task values were significantly greater than pre-task. In blue, we report results where pre-task values were greater than post-task. “None” refers to no significant differences between the two time points.

	CTL: Post-Task > Pre-Task	hBDI: Post-Task > Pre-Task
Theta	**ORIG: ***frontal & occipital* OSCIL: *none* **FRACT**: *most all electrodes*	**ORIG:** *frontal & occipital* OSCIL: *none* **FRACT**: *frontal & occipital*
Alpha	**ORIG:** *frontal & R parietal* **OSCIL:** *frontal & R occipital* **FRACT**: *most all electrodes*	ORIG: *none* OSCIL: *none* **FRACT**: *frontal*
Beta	**ORIG:** *from frontal to occipital* **OSCIL: ***central* **FRACT**: *frontal & occipital*	ORIG: *none* **OSCIL:** *R parieto-occipital* FRACT: *none*
Gamma	**ORIG:** *frontal & occipital* **OSCIL**: *frontal & occipital* **FRACT**: *most all electrodes*	**ORIG: ***L & R centro-parietal* **OSCIL**: *fronto-centro-parietal* **FRACT**: *L centro-parietal & R fronto-central*

## Data Availability

The data used in this project were obtained from the OpenNeuro database (https://openneuro.org, accessed on 3 March 2023). OpenNeuro is a data-sharing platform supported by the National Institutes of Health (NIH), the National Science Foundation (NSF), and other funding agencies. The original raw EEG recordings and clinical and demographic data can be downloaded at https://openneuro.org/datasets/ds003478/versions/1.1.0, accessed on 3 March 2023 [34]. The behavioral performance at the probabilistic learning tasks can be obtained by contacting Dr. James F. Cavanagh (jcavanagh@unm.edu), as indicated on OpenNeuro. The custom MATLAB-based code used to perform our data analyses can be provided upon reasonable request from the corresponding authors.

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
