# Peer review of "Resting-State EEG Alterations of Practice-Related Spectral Activity and Connectivity Patterns in Depression"

_biomedicines, 2024, doi:10.3390/biomedicines12092054_

Round 1

Reviewer 1 Report

Comments and Suggestions for Authors

This study is investigating the relationship between depression and changes in brain activity, specifically looking at energy regulation and neural plasticity. There are some issues that should be addressed:

Authors did not adjust for confounders such as age, gender, socioeconomic status, etc. in their comparisons.

Participants are college students (18-25 years old) and therefore, the results cannot generalized to other populations.

Why did the authors categorized the participants based on their BDI score? Why didn’t they use the continous data as it gives more power and more reliable results. Also, they excluded those with BDI score of between 7 and 16, which doesn’t make sense.

The results are based on a specific probabilistic learning task, which may not reflect general cognitive processes or be applicable to other tasks or settings.

The preprocessing steps, such as visual inspection and component rejection, may not completely eliminate artifacts, potentially affecting the quality of the EEG data.

Two-way ANOVA repeated measures analysis may be helpful in comparing the changes between the groups.

Author Response

Query 1. This study is investigating the relationship between depression and changes in brain activity, specifically looking at energy regulation and neural plasticity. There are some issues that should be addressed:

 Authors did not adjust for confounders such as age, gender, socioeconomic status, etc. in their comparisons.

Answer 1. We thank the reviewer for this observation. Unfortunately, the database retrieved from OpenNeuro did not include information about socioeconomic status, limiting our ability to control for this variable. As highlighted in Table 1, our analysis confirmed that the two groups did not significantly differ in age. However, we highlighted that the depression group (hBDI) had a higher proportion of women compared to the control group, a finding that is in line with the greater prevalence of depression in women observed worldwide. Importantly, we found no significant differences between men and women across all clinical test scores. Additionally, to account for potential gender effects in our logistic regression model, we conducted a supplementary analysis restricted to women only, which confirmed the BIS subscale as a good predictor of depression as described in Lines 235-239  (“Logistic regression analysis showed that the BIS score was a good predictor of acute depressive symptomatology (X2=27.51, p<0.001; Wald stat= 20.24, p<0.001; AUC=0.77; Accuracy=0.75; Sensitivity=0.56; Specificity=0.87), with greater sensitivity when the analysis was restricted to women (CTL: N=39; hBDI: N=33; X2=18.65, p<0.001; Wald stat= 13.49, p=0.0002; AUC=0.77; Accuracy=0.74; Sensitivity=0.70; Specificity=0.77).”)

In addition, we now discuss this point among the limitation of the study (see lines: 594-596: “Specifically, this dataset did not provide information about the socioeconomic status of the participants; did not include participants with a BDI score between 7 and 16 (thus precluding analysis with continuos BDI data” and lines 613-615: “Therefore, caution should be exercised in extending these results to older people or to populations from different socioeconomic and cultural backgrounds and future investigations are needed to address these issues.”)

Q2. Participants are college students (18-25 years old) and therefore, the results cannot generalized to other populations.

A2. We acknowledge the reviewer’s concern regarding the generalizability of our findings. Indeed, the used dataset consists of college students aged 18-25, which may limit the applicability of the results to broader populations. However, we believe that this age group provides a valuable model for studying subclinical depression and anxiety symptoms, given that it represents a critical period for the onset of mood disorders and cognitive changes. We have emphasized this limitation in the revised manuscript and discussed the need for future studies to validate our findings in more diverse and older populations for generalizability of our results (Lines 613-615: Therefore, caution should be exercised in extending these results to older people or to populations from different socioeconomic and cultural backgrounds and future investigations are needed to address these issues.).

Q3. Why did the authors categorized the participants based on their BDI score? Why didn’t they use the continous data as it gives more power and more reliable results. Also, they excluded those with BDI score of between 7 and 16, which doesn’t make sense.

A3. We thank the reviewer for pointing out this important aspect. The dataset we used was obtained from the OpenNeuro database, and it was already categorized based on the participants’ BDI scores. Specifically, the original study by Cavanagh and colleagues (2011), categorized participants into two groups: those with a BDI score below 7 and those with a BDI score above 16. Unfortunately, this meant that individuals with scores between 7 and 16 were excluded in the dataset we downloaded, and we were unable to access the raw continuous data to adjust this categorization.

Thus, while we acknowledge that analyzing continuous data can provide more statistical power and reliability, we were constrained by the structure of the available dataset. We now report this among the limitations of the study (see lines: 594-596: “Specifically, this dataset did not provide information about the socioeconomic status of the participants; did not include participants with a BDI score between 7 and 16 (thus precluding analysis with continuos BDI data”)

A4. The results are based on a specific probabilistic learning task, which may not reflect general cognitive processes or be applicable to other tasks or settings.

Q4. We appreciate the reviewer’s thoughtful comment. Indeed, the probabilistic learning task induces specific local changes thus limiting the topographical generalizability of our findings to other type of tasks. However, as discussed in the introduction, we wished to investigate general processes shared by different learning tasks that, anyway, share plasticity mechanisms and metabolic activities to sustain them. Thus, while the topographical results of this specific investigation applies to the employed task, we hypothesize that the pattern of changes in terms of frequency may be the same. Nevertheless, we now indicated in the limitations session that future research is needed to extend and replicate the present findings with different learning tasks. (See: Lines 615-618: Moreover, it would be important to replicate the present findings using other types of tasks, which may induce different topographical patterns, to assess the generalizability of our conclusions in terms of frequency changes.)

Q5. The preprocessing steps, such as visual inspection and component rejection, may not completely eliminate artifacts, potentially affecting the quality of the EEG data.

A5. We thank the reviewer for this insightful comment. We acknowledge that while our preprocessing steps, which are commonly employed in any standard EEG preprocessing pipeline, are designed to minimize artifacts, it is indeed possible that some residual artifacts may remain in the EEG data, affecting the signal-to-noise ratio. However, the residual artifacts are an inherent and accepted challenge in EEG research. In any event, we have taken great care to apply robust preprocessing techniques, including both visual inspection and component rejection, to ensure that the impact of any remaining noise is minimized. Importantly, our analyses of the fractal and pure oscillatory components of the signal, are robust to control the impact of EEG artifacts. This dual-component approach allows us to control for the impact of artifacts more effectively. By focusing on the pure oscillatory component, which is less prone to contamination by artifacts like muscle movements or external electrical noise, we are able to enhance the signal-to-noise ratio and obtain a more accurate and reliable understanding of the underlying neural dynamics.

Q6. Two-way ANOVA repeated measures analysis may be helpful in comparing the changes between the groups.

A6. We appreciate the reviewer’s suggestion. However, we would like to clarify that in our analysis, we employed FieldTrip’s functions—a widely used toolbox for EEG/MEG data analysis—which utilizes permutation-based non-parametric statistical testing. This approach is particularly well-suited for EEG data due to its flexibility in handling complex data structures – including topography-, its control for the multiple comparison problem due to the number of EEG electrodes, and its robustness against violations of assumptions like normality, which are often required in traditional parametric tests such as a two-way ANOVA repeated measures analysis. FieldTrip does not directly support or employ two-way ANOVA repeated measures analysis. Instead, it focuses on non-parametric testing methods that are more appropriate for the type of data we tested, in that topography must be taken into account. Finally, we wish to point out that, as reported in the statistical methods, we used a critical alpha of 0.05 and a minimum of two neighboring electrodes to test for the existence of a cluster of electrodes and thus, to produce information about topographical differences.

Reviewer 2 Report

Comments and Suggestions for Authors

The current manuscript by Elisa Tatti and her team aims to investigate the unique effects of depression and anxiety on EEG responses. Utilizing a publicly available EEG dataset, the authors report that at baseline, the hBDI group exhibited greater beta power in fronto-parietal regions and gamma power in the right parieto-occipital area compared to the CTL group. After practice, only the CTL group showed increases across all frequency ranges, suggesting that in depression, pre-existing high beta power may limit further growth post-task due to a possible inhibitory/excitatory imbalance and altered plasticity.

This research premise is novel, focusing on beta range spectral activity, a less-explored area in depression research. It examines dynamic changes in EEG power and connectivity in response to practice, providing insights into neuroplasticity deficits in depression, and integrates both spectral and connectivity analyses for a detailed understanding of neural alterations in depression.

The study design is well-conceived and appropriate for investigating the proposed research questions regarding depression and neurophysiological changes before and after a learning task. The manuscript is well-structured and cites recent and relevant references.

Overall, this study could significantly contribute to the existing literature by identifying potential biomarkers of impaired plasticity in depression and advancing the understanding of the disorder's neurophysiological basis.

I do not have any major concerns but do have a few suggestions:

1.      While the logistic regression model showing that BIS accurately identified 70% of individuals with high BDI scores is promising, the small sample size in each group necessitates caution.

2.      Although having 46 individuals in the hBDI group and 75 in the CTL group provides a balanced comparison, ideally, the groups should be as equal in size as possible.

These findings may not generalize to the broader population due to factors such as age, gender, socioeconomic status, and comorbid conditions. While the limitations have been identified, they should be clearly stated in the limitations section.

3.      If permissible by the grant, authors should deposit the custom MATLAB-based Fieldtrip Toolbox data analysis scripts in a public repository.

4.      Keeping word limitations for the introduction in mind, the authors should briefly introduce the concept of analyzing spectral activity across different frequency bands (theta, beta, gamma). This would help set up readers for the expected results and discussion, which currently lacks in the introduction and proposed hypothesis that specifically focuses on beta bands.

5.      Similarly, If the authors could elaborate on the term "practice" when it is first introduced, specifically on line 21 in the abstract and/or line 71 in the introduction, it would be beneficial for general readers.

Author Response

Comment: The current manuscript by Elisa Tatti and her team aims to investigate the unique effects of depression and anxiety on EEG responses. Utilizing a publicly available EEG dataset, the authors report that at baseline, the hBDI group exhibited greater beta power in fronto-parietal regions and gamma power in the right parieto-occipital area compared to the CTL group. After practice, only the CTL group showed increases across all frequency ranges, suggesting that in depression, pre-existing high beta power may limit further growth post-task due to a possible inhibitory/excitatory imbalance and altered plasticity.

This research premise is novel, focusing on beta range spectral activity, a less-explored area in depression research. It examines dynamic changes in EEG power and connectivity in response to practice, providing insights into neuroplasticity deficits in depression, and integrates both spectral and connectivity analyses for a detailed understanding of neural alterations in depression.

The study design is well-conceived and appropriate for investigating the proposed research questions regarding depression and neurophysiological changes before and after a learning task. The manuscript is well-structured and cites recent and relevant references.

Overall, this study could significantly contribute to the existing literature by identifying potential biomarkers of impaired plasticity in depression and advancing the understanding of the disorder's neurophysiological basis.

I do not have any major concerns but do have a few suggestions:

Q1.      While the logistic regression model showing that BIS accurately identified 70% of individuals with high BDI scores is promising, the small sample size in each group necessitates caution.

A1. We thank the reviewer for such positive remarks on our work. About the first point, we agree that the small sample size and the specific population selected for this study necessitate caution in interpreting these results. However, it is worth noting that our findings are consistent with existing literature on the relationship between BIS and depressive symptoms. This consistency with prior research lends additional credibility to our results, even within the context of a limited sample size. A caveat for the small sample have been added in the revised version (see lines 455-456: “Despite the small sample, our results are in agreement with previous accounts [52,53] that depression may be associated with an overactive inhibitory system and...”.)

Q2.      Although having 46 individuals in the hBDI group and 75 in the CTL group provides a balanced comparison, ideally, the groups should be as equal in size as possible.

A2. We thank the reviewer for this comment. While we agree that a more equal sample size could potentially enhance the statistical power of our comparisons, we want to emphasize that the method we employed, specifically the Monte Carlo permutation testing, is robust to differences in group sizes. Monte Carlo permutation testing involves randomly permuting the labels of the data points and calculating the test statistic for each permutation to generate a distribution under the null hypothesis. Given that Monte Carlo permutation testing does not require equal sample sizes, the imbalance between the 46 individuals in the hBDI group and the 75 in the CTL group does not compromise the validity of our results.

Q3. These findings may not generalize to the broader population due to factors such as age, gender, socioeconomic status, and comorbid conditions. While the limitations have been identified, they should be clearly stated in the limitations section.

A3. Thank you for this valuable consideration. We now clearly state this important aspect in our limitation section. (See lines 594-596: “Specifically, this dataset did not provide information about the socioeconomic status of the participants; did not include participants with a BDI score between 7 and 16 (thus precluding analysis with continuous BDI scores”); and lines 610-618: “Another feature of this investigation is that subjects were all college students within a narrow age range. This aspect likely decreased variability but makes it critical to validate the present results for other age ranges and populations with stressors other than those related to specific life experiences. Therefore, caution should be exercised in extending these results to older people or to populations from different socioeconomic and cultural backgrounds and future investigations are needed to address these issues. Moreover, it would be important to replicate the present findings using other types of tasks, which may induce different topographical patterns, to assess the generalizability of our conclusions in terms of frequency changes.”)

Q4.      If permissible by the grant, authors should deposit the custom MATLAB-based Fieldtrip Toolbox data analysis scripts in a public repository.

A4. Thank you for suggesting that we share the data analysis code. While we fully recognize the importance of transparency and reproducibility in scientific research, we have decided to provide the codes upon reasonable request due to intellectual property concerns. We have added this opportunity in the Data Availability Statement. Moreover, to ensure that our analysis can be understood and replicated, we have included a comprehensive description of the algorithms and methodologies used in the study, along with the specific names of the functions employed (see methods: 2.2 EEG data acquisition and preprocessing). Additionally, all data necessary to replicate the analysis are freely available on OpenNeuro.

Q5.      Keeping word limitations for the introduction in mind, the authors should briefly introduce the concept of analyzing spectral activity across different frequency bands (theta, beta, gamma). This would help set up readers for the expected results and discussion, which currently lacks in the introduction and proposed hypothesis that specifically focuses on beta bands.

A5. We appreciate your suggestion to briefly introduce the concept of analyzing spectral activity across different frequency bands. In our revised manuscript, we added a brief overview of spectral activity in different frequency bands in depression, a topic that anyway is expanded in the discussion. (see: Lines 44-51): “Power analysis of EEG frequency bands in individuals with depression can yield in-sightful results by disclosing alterations that occur in specific bands and that may be associated with neural mechanisms underlying depressive symptoms. Notably, a de-bated EEG correlate of major depression is the frontal asymmetry in the alpha range, that is, the unequal activation of the two hemispheres [3] often associated with altered emo-tional processing and regulation. Moreover, different degrees and patterns of alterations in beta [4] and gamma [5] bands have been reported in depression.”)

Q6.      Similarly, If the authors could elaborate on the term "practice" when it is first introduced, specifically on line 21 in the abstract and/or line 71 in the introduction, it would be beneficial for general readers.

A6. We appreciate the reviewer’s suggestion to clarify the term "practice" for general readers. In response, we have added clarifying terms to specify the type of practice we are referring to in the abstract (lines 21 and 24) and introduction (line 74), to make the context clearer and more accessible to the readers.